# Active Approximately Metric-Fair Learning

**Yiting Cao**[1]                              **Chao Lan**[1]

[1]School of Computer Science, University of Oklahoma, Norman, Oklahoma, USA

## Abstract

Existing studies on individual fairness focus on the passive setting and typically require $O(\frac{1}{\varepsilon^2})$ labeled instances to achieve an $\varepsilon$ bias budget. In this paper, we build on the elegant Approximately Metric-Fair (AMF) learning framework and propose an active AMF learner that can provably achieve the same budget with only $O(\log \frac{1}{\varepsilon})$ labeled instances. To our knowledge, this is a first and substantial improvement of the existing sample complexity for achieving individual fairness. Through experiments on three data sets, we show the proposed active AMF learner improves fairness on linear and non-linear models more efficiently than its passive counterpart as well as state-of-the-art active learners, while maintaining a comparable accuracy. To facilitate algorithm design and analysis, we also design a provably equivalent form of the approximate metric fairness based on uniform continuity instead of the existing almost Lipschitz continuity.

## 1 INTRODUCTION

In modern machine learning applications, an important societal concern is fairness of the machine-learned models. One may think that model-made decisions have no discrimination against minority people, but many case studies show otherwise e.g., Feller et al. [2016], Chan and Wang [2018]. As a result, how to attain fairness in machine-learned models becomes an important research problem and the literature has exploded in recent years. See e.g., some latest surveys Chouldechova and Roth [2018], Mehrabi et al. [2021], Pessach and Shmueli [2022] and references therein.

Model fairness has been studied at both group level and individual level. Roughly speaking, group fairness requires model outputs to have small disparity across different groups of people, while individual fairness requires model outputs

to be similar on similar individuals. This paper focuses on individual fairness.

Individual fairness was initially formalized as the Lipschitz condition of a prediction model Dwork et al. [2012], and later relaxed to a probabilistic and almost Lipschitz condition called approximate metric-fairness Yona and Rothblum [2018]. There are many studies on different aspects of individual fairness such as how to design the fairness metric Ilvento [2020], Mukherjee et al. [2020], how to achieve fairness with limited resources Kim et al. [2018], Bechavod et al. [2020], and how to combine individual fairness with group fairness Zemel et al. [2013], Sharifi-Malvajerdi et al. [2019]. The sample complexity for achieving individual fairness in passive learning is studied in Balashankar and Lees [2019], Shabat et al. [2020].

This paper studies a new aspect of individual fairness. We ask *can one obtain a more efficient sample complexity for achieving individual fairness through active learning?* To our knowledge, all prior studies focus on the passive setting and maintain an $O(\frac{1}{\varepsilon^2})$ sample complexity for bounding (properly defined) individual bias by $\varepsilon$. See Yona and Rothblum [2018], Balashankar and Lees [2019], Shabat et al. [2020] for example. In this paper, we show it can be improved to $O(\log \frac{1}{\varepsilon})$ through active learning.

To facilitate algorithm design and analysis, we first present a new form of approximate metric-fairness (AMF) based on uniform continuity and prove its equivalence to the original form using an interesting connection between uniform continuity and *almost* Lipschitz continuity Vanderbei [1991]. Based on this, we present a passive AMF learner and prove the generalization ability of its achieve fairness.

Building on the above, we design an active AMF learner which labels instances that are fairly close to their neighbors but receive fairly different predictions. Under proper conditions, we prove this learner only takes $O(\log \frac{1}{\varepsilon})$ labeling to bound the bias of its returned model by $\varepsilon$ with high probability. Our analysis assumes boundness of a specially designed counter approximate metric-fairness coefficient,

*Accepted for the 38th Conference on Uncertainty in Artificial Intelligence* (UAI 2022).

and we exemplify the calculation of it.

At the end, we experiment the proposed active AMF learner on three real-world data sets. We observe it improves individual fairness of both linear and non-linear models more efficiently than its passive counterpart as well as state-of-the-art active learners while maintaining a comparable accuracy, achieving a more efficient fairness-accuracy trade-off.

The rest of this paper is organized as follows. We review related topics in Section 2 and present the proposed AMF form and passive learner in Section 3. In section 4, we present the proposed active AMF learner and prove its sample complexity. We discuss several implementation issues in Section 5, present experimental results in Section 6, and conclude the study in Section 7. Proofs of certain theoretical results are elaborated in the supplementary material.

## 2    BACKGROUND

### 2.1    FAIRNESS IN MACHINE LEARNING

Today, machine-learned models are widely used in sensitive domains like healthcare and hiring, and it is imperative for them to give fair assessment on human candidates. Take hiring as an example, when a model is used to score the qualification of job candidates, for fairness it should give similar scores to similar candidates disregarding their race or gender (i.e. no racial or gender discrimination). In reality, however, many model assessments are considered unfair Feller et al. [2016], Chan and Wang [2018]. This has motivated intensive research on how to attain fairness in machine-learned models e.g. Grgic-Hlaca et al. [2016], Alabi et al. [2018], Grgić-Hlača et al. [2018], Rothblum and Yona [2018], Mozannar et al. [2020], to name a few.

Model fairness has been studied at both group and individual levels Dwork et al. [2012]. In this paper, we focus on individual fairness which, roughly speaking, requires model output to be similar on similar individuals.

Individual fairness is first formalized as the Lipschitz condition of the model Dwork et al. [2012], and later relaxed to a probabilistic and almost Lipschitz condition called Approximate Metric-Fairness (AMF) Yona and Rothblum [2018][1] Many later studies are built on AMF Balashankar and Lees [2019], Bechavod et al. [2020], Kim et al. [2018]. The active fair learner proposed in this paper is also built on AMF, but we present a new and provably equivalent form based on uniform continuity instead of almost Lipschitz.

One research direction in individual fairness is to attain a proper metric for evaluating individual similarity Ilvento [2020], Mukherjee et al. [2020]. In this paper, we assume a metric is given and focus on how to achieve individual

---

[1]In many following discussions, we will use AMF to represent 'approximate metric-fairness' or 'approximately metric-fair'.

fairness *efficiently* through active learning.

Finally, to our knowledge, existing studies on individual fairness focus on the passive setting, where training data are randomly labeled. Their typical sample complexity for achieving individual fairness is $O(\frac{1}{\varepsilon^2})$ Yona and Rothblum [2018], Balashankar and Lees [2019], Shabat et al. [2020]. In this paper, we focus on the active setting, where training data are strategically labeled. We show the proposed active AMF learner admits an $O(\log \frac{1}{\varepsilon})$ sample complexity, which substantially improves the state-of-the-art result.

### 2.2    ACTIVE LEARNING

Active learning has been extensively studied in the literature Settles [2009], Aggarwal et al. [2014], Hanneke et al. [2014]. Given a supervised learner, active learning assumes labels of the training data are expensive to query and aims to minimize the query cost by strategically labeling a few data for efficiently improving model accuracy. For example, the uncertainty-based strategy labels data with uncertain model predictions, and the query-by-committee strategy labels data receiving disagreed predictions from a committee of models. Active labeling strategies have been successfully applied in many domains and shown to improve model accuracy more efficiently than random labeling Thompson et al. [1999], Warmuth et al. [2003], Liu [2004], Hoi et al. [2006], Abe et al. [2006], Zhao and Hoi [2013].

On the theory side, active labeling can allows one to learn a model with $\varepsilon$ error by labeling only $O(\log \frac{1}{\varepsilon})$ instances, and this is more efficient than random labeling which requires $O(\frac{1}{\varepsilon})$ instances to achieve the same error guarantee Dasgupta [2005], Hanneke [2007], Balcan et al. [2010].

We notice that most active labeling strategies are designed for classification model but very few are for regression model Burbidge et al. [2007], Sugiyama and Nakajima [2009], Cai et al. [2013], Yu and Kim [2010]. To our knowledge, the state-of-the-art strategy for regression model is greedy sampling Wu et al. [2019], which labels data that are most different from the already labeled training data in both feature space and label space.

Our study is related to active learning but differs in that they focus on improving accuracy for traditional learners, while we focus on improving individual fairness for AMF learners. Despite the difference, our work is inspired by disagreement-based active learning Hanneke [2007].

### 2.3    FAIRNESS IN ACTIVE LEARNING

The intersection of fairness and active learning is a fairly new research direction, and existing studies can be roughly grouped into active labeling Anahideh et al. [2020], Sharaf and Daumé III [2020] and adaptive sampling Abernethy et al. [2020], Shekhar et al. [2021]. This paper considers the

active labeling setting, but differs from the existing study as they focus on improving group fairness for standard learner while we focus on improving individual fairness for AMF learner. Besides, we are the first work that shows active learning can improve the sample complexity for individual fairness to $O(\log \frac{1}{\varepsilon})$, which is not presented in prior studies.

# 3 APPROXIMATE METRIC-FAIRNESS

In this section, we present a new form of approximate metric-fairness (AMF) and prove its equivalence to the original form. Then we present the corresponding passive AMF learner and prove its generalization guarantee. Our study is focused on the regression problem.

Let $X$ be an instance space equipped with a metric $d$ and distribution $D$. Let $H$ be a class of models defined on $X$. The original form of AMF Yona and Rothblum [2018] is defined based on almost Lipschitz continuity, as follows.

**Definition 3.1.** A model $h \in H$ is said to be $(\varepsilon, \beta)$ approximately metric-fair with respect to $d$ and $D$ if

$$\Pr_{x,x'\sim D}\{|h(x) - h(x')| > d(x, x') + \beta\} \leq \varepsilon. \quad (1)$$

To facilitate algorithm design and analysis, we propose the following new form of AMF based on uniform continuity.

**Definition 3.2.** A model $h \in H$ is said to be $(\alpha, \beta, \varepsilon)$ approximately metric-fair with respect to $d$ and $D$ if

$$\Pr_{x,x'\sim D}\{d(x, x') \leq \alpha, |h(x) - h(x')| > \beta\} \leq \varepsilon. \quad (2)$$

Intuitively, the new form models individual fairness by stating that if two individuals $x$ and $x'$ are similar (in a sense that $d(x, x') \leq \alpha$), then their predictions should be similar (in a sense that $|h(x) - h(x')| > \beta$) with high probability.

Our following theorem suggests the two forms are equivalent, and is inspired by an interesting discovery that uniform continuity is *almost* Lipschitz Vanderbei [1991].

**Theorem 3.3.** *Fix any $\alpha, \beta > 0$. Any model with a convex domain is $(\varepsilon, \beta)$ approximately metric-fair with respect to $d$ and $D$ if it is $(\alpha, \beta, \varepsilon)$ approximately metric-fair with respect to metric $d' = \frac{\alpha}{2\beta} \cdot d$ and $D$, and only if it is $(\alpha, 3\beta, \varepsilon)$ approximately metric-fair with respect to $d'$ and $D$.*

*Proof.* Let $h$ be a model with a convex domain. Define two sets $\Psi_1(\beta) = \{(x, x') \mid |h(x) - h(x')| \leq d(x, x') + \beta\}$ and $\Psi_2(\alpha, \beta) = \{(x, x') \mid d'(x, x') \leq \alpha \Rightarrow |h(x) - h(x')| \leq \beta\}$, where '$\Rightarrow$' means 'imply'. We first prove

$$\Psi_2(\alpha, \beta) \subseteq \Psi_1(\beta) \subseteq \Psi_2(\alpha, 3\beta). \quad (3)$$

The left relation holds because, for any $\beta$, if there exists an $\alpha$ such that $d'(x, x') \leq \alpha$ implies $|h(x) - h(x')| \leq \beta$, then

[Vanderbei, 1991, Theorem 1] implies that $|h(x) - h(x')| \leq d(x, x') + \beta$, where $d(x, x') = \frac{2\beta}{\alpha}d'(x, x')$.

The right relation follows from the fact that, if $|h(x) - h(x')| \leq d(x, x') + \beta = \frac{2\beta}{\alpha}d'(x, x') + \beta$, then $d'(x, x') \leq \alpha$ implies $|h(x) - h(x')| \leq \frac{2\beta}{\alpha} \cdot \alpha + \beta = 3\beta$.

Then, by contrapositive, (3) implies

$$\tilde{\Psi}_2(\alpha, \beta) \supseteq \tilde{\Psi}_1(\beta) \supseteq \tilde{\Psi}_2(\alpha, 3\beta), \quad (4)$$

where $\tilde{\Psi}$ denotes the complement of $\Psi$. This further implies $\Pr\{\tilde{\Psi}_2(\alpha, \beta)\} \geq \Pr\{\tilde{\Psi}_1(\beta)\} \geq \Pr\{\tilde{\Psi}_2(\alpha, 3\beta)\}$, and the theorem follows by the two definitions of fairness. $\square$

Theorem 3.3 shows one form of AMF converges to the other as $\beta$ decreases, which establishes an equivalence between them. It also suggests we can achieve one form of AMF through the other. In the rest of this paper, we will design and analyze AMF learners based on Definition 3.2. For conciseness, we will omit the subscripts in $\Pr_{x,x'\sim D}$ whenever they are clear from the context.

## 3.1 AMF LEARNING WITH PERFORMANCE GENERALIZATION GUARANTEE

In this section, we present a passive AMF learner based on Definition 3.2 and prove its generalization guarantee.

To facilitate discussion, define the fairness measure

$$\Delta_{\alpha,\beta}(h) = \Pr\{d(x, x') \leq \alpha, |h(x) - h(x')| > \beta\}. \quad (5)$$

Then $h$ is said to be $(\alpha, \beta, \varepsilon)$-AMF if $\Delta_{\alpha,\beta}(h) \leq \varepsilon$.

Let $S$ be a sample of $X \times X$ with cardinality $m$. An estimate of the probability $\Delta_{\alpha,\beta}(h)$ on sample $S$ is

$$\Delta_{\alpha,\beta}(h; S) = \frac{1}{m} \sum_{(x,x')\in S} \mathbb{I}\{d(x, x') \leq \alpha, \\ |h(x) - h(x')| > \beta\}, \quad (6)$$

where $\mathbb{I}$ is an indicator function.

It is natural for AMF learning to find a model $h$ with small $\Delta_{\alpha,\beta}(h; S)$ and hope this could generalize to a small $\Delta_{\alpha,\beta}(h)$. In this paper, we focus on a realizable case where $H$ contains perfect AMF models that satisfy $\Delta_{\alpha,\beta}(h) = 0$. Based on this, we define the passive AMF learner as follows.

**Definition 3.4.** Given a hypothesis class $H$, a loss function $\ell$ and a labeled training set $L = \{(x_1, y_n), \dots, (x_n, y_n)\}$ where $x_i$ is the $i_{th}$ instance and $y_i$ is its label, an AMF learner returns a model $h \in H$ by solving

$$\min_{h \in H} \frac{1}{n} \sum_{i=1}^{n} \ell(h(x_i), y_i), \quad \text{s.t. } \Delta_{\alpha,\beta}(h; S) = 0, \quad (7)$$

where $S = \{(x_i, x_j)\}_{i,j=1,\dots,n}$.

We can show the above AMF learner has a similar generalization guarantee as in Yona and Rothblum [2018] based on the following lemma. Let $\mathcal{R}_m(\cdot)$ denote the Rademacher complexity of some hypothesis class for sample size $m$.

**Lemma 3.5.** *Fix any $t, \beta > 0$. Let $F : X \times X \to \mathbb{R}$ be a hypothesis class induced from $H$ such that $\forall f \in F$, $f(x, x') = \tau_\beta^t(|h(x) - h(x')|)$ where $\tau_\beta^t(z)$ is a piecewise model outputting 1 if $z > \beta + \frac{1}{t}$, outputting 0 if $z \leq \beta$ and $t(z - \beta)$ otherwise. Then $\mathcal{R}_m(F) \leq 8t \cdot \mathcal{R}_m(H)$.*

*Proof Sketch.* Repeatedly apply the Rademacher complexity property of composite function with Lipschitz condition e.g. [Bartlett and Mendelson, 2002, Theorem 12] on $\tau_\beta^t$ and $abs$. See the supplementary material for details. $\square$

Based on the above, we can prove the proposed AMF learner has generalization guarantee based on an assumption that instances are sampled i.i.d.. The results is as follows.

**Theorem 3.6.** *Fix any $\alpha, \beta, t > 0$. Suppose $\mathcal{R}_m(H) \in O(1/\sqrt{m})$. Any model $h \in H$ returned by the AMF learner satisfies $\Delta_{\alpha, \beta+1/t}(h) \leq \varepsilon$ with probability at least $1 - \delta$ if $m \geq \frac{1}{\varepsilon^2} \left( 16tc + \sqrt{\frac{1}{2} \log \frac{1}{\delta}} \right)$, where $m$ is the number of $(x, x') \in S$ satisfying $d(x, x') \leq \alpha$ and $c$ is a constant inherited from $O(1/\sqrt{m})$.*

*Proof Sketch.* The main challenge in our analysis is an extra $d(x, x') \leq \alpha$ term that cannot be directly removed using the Rademacher complexity property as in Lemma 3.5. To tackle this, we introduce $V = \{(x, x') \in S; d(x, x') \leq \alpha\}$.

We will first transform the analysis of joint event $|h(a) - h(b)| > \beta$ and $d(a, b) \leq$ to an analysis of single event $|h(a) - h(b)| > \beta$ by narrowing the domain to $V$. Then, we derive a generalization bound for the single event by first relaxing its indicator function to the piecewise function defined in Lemma 3.5, then applying the standard generalization argument with $\mathcal{R}_m(F)$ e.g., Mohri et al. [2018], and finally connecting $\mathcal{R}_m(F)$ to $\mathcal{R}_m(H)$ using Lemma 3.5. At the end, we transform the result for the single event back to a result for the joint event which completes the proof. See the supplementary material for details. $\square$

Theorem 3.6 implies one can achieve $(\alpha, \beta, \varepsilon)$ AMF with $O(\frac{1}{\varepsilon^2})$ randomly labeled instances, which is consistent with the sample complexity in Yona and Rothblum [2018]. Constant $c$ depends on the hypothesis class e.g., if $H$ is the set of linear models with proper constraints, we can set $c$ to the maximum norm of the instance Shalev-Shwartz and Ben-David [2014]; if $H$ is the set of kernel machines with proper constraints, we can set $c$ to the product of kernel function bound and gram matrix trace Mohri et al. [2018].

In the theorem, variable $t$ is the slope of a Lipschitz function introduced to approximate the indicator function. Its impact

---

**Algorithm 1** Active AMF Learning

**Input:** an initial labeled training set $L$, an unlabeled set $U$, a hypothesis class $H$, number $k$.
1: **while** stopping criterion is not met **do**
2:     Learn a model $h \in H$ based on sample $L$ using the AMF learner in Definition 3.4.
3:     Pick an i.i.d. sample of $k$ instances $u \in U$ satisfying

$$\exists u' \in L, \ d(u, u') \leq \alpha, \ |h(u) - h(u')| > \beta. \quad (8)$$

4:     Label the selected instances. Then add them to sample $L$, and remove them from sample $U$.
5: **end while**
**Output:** model $h$.

---

on the error bound is interesting twofold. A smaller $t$ leads to a weaker fairness guarantee, in a sense that $\Delta_{\alpha, \beta+1/t'} \leq \varepsilon$ implies $\Delta_{\alpha, \beta+1/t} \leq \varepsilon$ whenever $t' \leq t$. But it also leads to higher sample efficiency, in a sense that a smaller $t$ implies smaller $m$ suffices for the generalization guarantee.

## 4 ACTIVE AMF LEARNING

In this section, we propose an active AMF learner based on Definition 3.4 and derive its sample complexity.

Our key idea is to label instances that are fairly close to their neighbors but receive fairly different predictions from some hypothesis. We characterize such instances using a set

$$\mathcal{C}_{\alpha, \beta}(H) = \{(x, x') \in X \times X; \exists h \in H, \\ d(x, x') \leq \alpha, \ |h(x) - h(x')| > \beta\}. \quad (9)$$

Next, we design a counter AMF coefficient, which will be used to derive the complexity.

**Definition 4.1.** The counter $(\alpha, \beta)$ AMF coefficient with respect to a hypothesis class $H$ is

$$\xi_{\alpha, \beta} = \sup_{r > 0} \frac{\Pr\{(x, x') \in \mathcal{C}_{\alpha, \beta}(\mathcal{B}_{\alpha, \beta}(r))\}}{r}, \quad (10)$$

where $\mathcal{B}_{\alpha, \beta}(r) = \{h \in H; \Delta_{\alpha, \beta}(h) \leq r\}$ is the set of hypotheses that are $(\alpha, \beta, r)$ AMF.

Intuitively, the coefficient measures the largest volume of instance pairs that do not contribute to the fairness achievable in a hypothesis class. We could expect it to be smaller if hypotheses are more fair. For conciseness, we will omit the subscripts in $\xi_{\alpha, \beta}$ whenever they are clear from the context.

The proposed active AMF learner is shown in Algorithm 1. In each round, it trains model $h$ on the labeled set using the AMF learner, and then labels instances that are close to the training data but receive different predictions from $h$. It is clear that all labeled instances fall in $\mathcal{C}_{\alpha, \beta}(H)$. The fairness

coefficients $\alpha, \beta$ are assumed preset by the problem, and we can stop labeling when a desired AMF degree is achieved.

Our following theorem shows that, under proper conditions, Algorithm 1 can return a model satisfying $(\alpha, \beta, \varepsilon)$ AMF through $O(\log \frac{1}{\varepsilon})$ labeling with high probability.

**Theorem 4.2.** *Fix any $\alpha, \beta > 0$. If the counter $(\alpha, \beta)$ AMF coefficient w.r.t. $H$ is bounded, then with probability at least $1 - \delta$, any $h \in H$ returned by Algorithm 1 satisfies $\Delta_{\alpha,\beta}(h) \leq \varepsilon$ after $O(\log \frac{1}{\varepsilon})$ labeling.*

*Proof Sketch.* Let $V_q = \{h \in H; \Delta_{\alpha,\beta}(h; S_q) = 0\}$ be the set of 'perfect' AMF models at the end of $q$ rounds of labeling. The goal of our analysis is to show that, if we label $k = \frac{1}{4\xi^2}\left(32c/\beta + \sqrt{\frac{1}{2}\log\frac{1}{\delta'}}\right)$ instances in each round, then by the generalization bound in Theorem 3.6, there is

$$\Pr\{\mathcal{C}_{\alpha,\beta}(V_{q+1})\} \leq \frac{1}{2}\Pr\{\mathcal{C}_{\alpha,\beta}(V_q)\}. \quad (11)$$

with high probability. This implies $Q = \log_2\frac{1}{\varepsilon}$ rounds of labeling, which means $Qk \in O(\log\frac{1}{\varepsilon})$ total labeling, suffices for $\Pr\{\mathcal{C}_{\alpha,\beta}(V_{q+1})\} \leq \varepsilon$. Since $\Delta_{\alpha,\beta}(h) \leq \Pr\{\mathcal{C}_{\alpha,\beta}(V_q)\}$ for any $h \in V_q$ by definition, the theorem is proved.

Let $\&$ be logic 'AND' and define event

$$I_\alpha^\beta(x, x'; h) := d(x, x') \leq \alpha \ \& \ |h(x) - h(x')| > \beta. \quad (12)$$

A key to prove (11) is to split the domain of $\Delta_{\alpha,\beta}(h) = \Pr\{I_\alpha^\beta(x, x'; h)\}$ for any $h \in V_{q+1}$ into $(x, x') \in \mathcal{C}_{\alpha,\beta}(V_q)$ and $(x, x') \notin \mathcal{C}_{\alpha,\beta}(V_q)$. Probability on the second subdomain is zero, and probability on the first subdomain can be bounded using Theorem 3.6 conditioned on the fact that all labeled instances fall in $\mathcal{C}_{\alpha,\beta}(V_q)$. That bound is smaller than $\frac{1}{2\xi}$ by our choice of $k$ and the definition of $\xi$, therefore implying $V_{q+1} \subseteq \mathcal{B}\left(\frac{\Pr\{\mathcal{C}_{\alpha,\beta}(V_q)\}}{2\xi}\right)$ and thus $\Pr\{\mathcal{C}_{\alpha,\beta}(V_{q+1})\} \leq \Pr\left\{\mathcal{C}_{\alpha,\beta}\left(\mathcal{B}_{\alpha,\beta}\left(\frac{\Pr\{\mathcal{C}_{\alpha,\beta}(V_q)\}}{2\xi}\right)\right)\right\} \leq \xi \cdot \frac{\Pr\{\mathcal{C}_{\alpha,\beta}(V_q)\}}{2\xi} = \frac{\Pr\{\mathcal{C}_{\alpha,\beta}(V_q)\}}{2}$, where the second inequality is by definition. This proves (11) and thus the theorem. $\square$

The proof of Theorem 4.2 also illuminates the key for Algorithm 1 to reduce labeled instances is in Step 3, where we label $u$ if $(u, u') \in \mathcal{C}_{\alpha,\beta}(V_q)$ because only such pair can be used to further rule out hypotheses in $V_q$ and shrink $\mathcal{C}_{\alpha,\beta}(V_q)$, which guarantees the shrinkage of $\Delta_{\alpha,\beta}(h)$.

We should mention an implicit assumption of the derived sample complexity is that, the unlabeled set contains at least one instance satisfying (8) per epoch until convergence. This is similar to the analysis of disagreement-based active learning Hanneke et al. [2014], which assumes at least one unlabeled instance is disagreed by the committee models per epoch. From a practical perspective, when no valid instance is found, we could train another model or randomly label one instance and proceed to the next epoch.

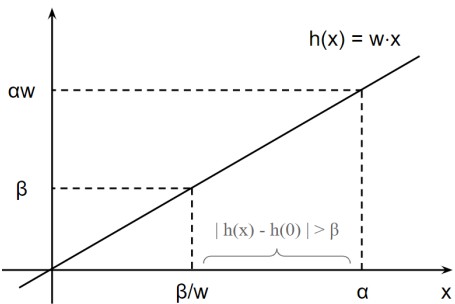

Figure 1: Visualization of $\Pr_x(I_\alpha^\beta(x, 0; h))$.

We should also mention the time complexity for Algorithm 1 to find an instance satisfying (8). In a centralized computing environment, the complexity is $O(|U||L|)$, where $|U|$ is the size of unlabeled set and $|L|$ is the size of labeled set. Typically $|L| \ll |U|$. This is higher than the complexity of uncertainty-based strategy which is typically $O(|U|)$, but more comparable to the complexity of query-by-committee which is typically $O(|U|t)$ for $t$ committee models. In a distributed computing environment, the complexity can be reduced to $O(|U|)$ if the evaluations of an instance $u \in U$ with all $u' \in L$ can be parallelized. Nonetheless, how to make selection more efficient remains an open challenge.

## 4.1 THE COUNTER AMF COEFFICIENT

An important factor in our analysis is the counter AMF coefficient. We give an example on how to calcualte it.

**Example 4.3.** Fix $\alpha, \beta, B > 0$. Let $h_w(x) = w \cdot x$ be a 1-dimensional linear hypothesis defined on $[-B, B]$, and define $H = \{h_w; w \geq 0\}$. Assume instances are uniformly distributed on $[-B, B]$. If $B > \alpha$, then

$$\Delta_{\alpha,\beta}(h) = \begin{cases} 0, & \text{if } w < \frac{\beta}{\alpha} \\ 1 - \frac{\beta}{\alpha w} + \frac{\beta}{Bw}\ln\frac{\beta}{\alpha w}, & \text{if } w \geq \frac{\beta}{\alpha} \end{cases}, \quad (13)$$

and the counter $(\alpha, \beta)$ AMF coefficient w.r.t. to $H$ is 1.

*Proof.* The roadmap of this proof is as follows. We will first derive $\Delta_{\alpha,\beta}(h_w)$ through case study and show it is non-decreasing with respect to $w$. Based on this, we will then argue the probability in (10) is equivalent to

$$\mathbb{P}_* := \Pr\{|h_*(x) - h_*(x')| > \beta, d(x, x') \leq \alpha\}, \quad (14)$$

where $h_*$ is the model satisfying $\Delta_{\alpha,\beta}(h_*) = r$, hence $\xi = \sup_{r>0} \mathbb{P}_*/r = \sup_{r>0} r/r = 1$.

Now we show the detailed proof. For conciseness, we will write $h$ for $h_w$ but with the mind that each $h$ is associated with a $w$. Also, recall the event notation $I_\alpha^\beta(x, x'; h) := d(x, x') \leq \alpha \ \& \ |h(x) - h(x')| > \beta$.

*Step 1: Characterize $\Delta_{\alpha,\beta}(h)$ for any $h \in H$.*

Fix any $h$. Consider two cases.

(i) If $\alpha w < \beta$, simple geometric analysis shows that event $I_\alpha^\beta(x, x'; h)$ is always false so $\Pr_{x,x'}\{I_\alpha^\beta(x, x'; h)\} = 0$.

(ii) If $\alpha w \geq \beta$ (which implies $w \neq 0$), then $\alpha \geq \beta/w$. In this case, we can properly partition the domain and have

$$\Pr_{x,x'}\{I_\alpha^\beta(x, x'; h)\}$$

$$= \mathbb{E}_{x' \in [-B,B]}\left[\Pr_x\{I_\alpha^\beta(x, x'; h)\}\right]$$

$$= 2\,\mathbb{E}_{x' \in [0,B]}\left[\Pr_x\{I_\alpha^\beta(x, x'; h)\}\right]$$

$$= 2 \int_{x' \in [0, B-\alpha]} \Pr_x\{I_\alpha^\beta(x, x'; h)\} \cdot p(x') \tag{15}$$

$$+ 2 \int_{x' \in (B-\alpha, B-\frac{\beta}{w}]} \Pr_x\{I_\alpha^\beta(x, x'; h)\} \cdot p(x')$$

$$+ 2 \int_{x' \in (B-\frac{\beta}{w}, B]} \Pr_x\{I_\alpha^\beta(x, x'; h)\} \cdot p(x'),$$

where $\Pr_x\{I_\alpha^\beta(x, x'; h)\}$ is the probability defined for $x$ with a fixed $x'$. In (15), the first equality is by definition, and the second equality is by the observation that $\Pr_x\{I_\alpha^\beta(x, x'; h)\}$ is symmetric on $[-B, B]$ (which will become more clear in later analysis). Note that $p(x') = \frac{1}{2B}$.

Now we study each integral separately.

(ii.a) If $x' \in [0, B-\alpha]$, we can show

$$\Pr_x\{I_\alpha^\beta(x, x'; h)\} = 1 - \frac{\beta}{w\alpha}. \tag{16}$$

To verify this, let us first fix $x' = 0$ and identify the set of $x$ in $[-\alpha, \alpha]$ that makes event $I_\alpha^\beta(x, 0; h)$ true. This case is illustrated in Figure 1. We see all targeted $x$ fall in $[\beta/w, \alpha]$ and (by symmetry) in $[-\alpha, -\beta/w]$. This implies $\Pr_x\{I_\alpha^\beta(x, 0; h)\} = \frac{2 \cdot (\alpha - \beta/w)}{2 \cdot \alpha} = 1 - \frac{\beta}{\alpha w}$. Since $h$ is linear, the above result applies to all $x' \in [0, B-\alpha]$, which implies (16) and thus the first integral equals to $(1 - \frac{\beta}{w\alpha})(1 - \frac{\alpha}{B})$. Note it is non-negative since $\alpha w \geq \beta$ and $B > \alpha$.

(ii.b) If $x' \in (B-\alpha, B-\frac{\beta}{w}]$, we can show

$$\Pr_x\{I_\alpha^\beta(x, x'; h)\} = 1 - \frac{\beta}{w(B - x')}. \tag{17}$$

We can verify this in a similar way as in (ii.a), with additional shift of the origin to $x'$ and constraint $x \leq B$. Then, geometric analysis suggests all targeted $x$ fall in $[x' + \frac{\beta}{w}, B]$ (shorter than interval $[\beta/w, \alpha]$ in Figure 1) and thus $\Pr_x\{I_\alpha^\beta(x, x'; h)\} = \frac{2(B - x' - \beta/w)}{2(B - x')}$. This implies the second integral is $\frac{1}{B}(\alpha - \frac{\beta}{w}(1 - \ln\frac{\beta}{w\alpha}))$. Note it is non-negative as (17) is non-negative by the domain of $x'$.

(ii.c) If $x' \in (B - \frac{\beta}{w}, B]$, it is easy to see no $(x', x)$ makes event $I_\alpha^\beta(x, x'; h)$ true so $\Pr_x\{I_\alpha^\beta(x, x'; h)\} = 0$. Then the third integral is zero.

Plugging the integrals of (ii.a), (ii.b) and (ii.c) back to (15), and combining results of cases (i) and (ii) gives (13).

*Step 2: Show $\Delta_{\alpha,\beta}(h)$ is non-decreasing w.r.t. $w$.*

All we need to show is $\Delta_{\alpha,\beta}(h)$ is non-negative and non-decreasing when $w \geq \frac{\beta}{\alpha}$. The first property is guaranteed since all integrals in (15) are non-negative. To see the second property, take derivative $\frac{\partial \Delta_{\alpha,\beta}(h)}{\partial w} = \frac{\beta\,(B + \alpha(\ln\frac{\alpha w}{\beta} - 1))}{w^2\,\alpha\,B}$. Since $w \geq \frac{\beta}{\alpha}$ and $B > \alpha$, we can easily show the derivative is bigger than zero and hence $\Delta_{\alpha,\beta}(h)$ is non-decreasing.

*Step 3: Equivalent Probability.*

Let $h_* = w_* x$ be the model satisfying $\Delta_{\alpha,\beta}(h_*) = r$. It is not hard to show it exists for every $r \in [0, 1)$ based on (13). Then, results of Step 1 and Step 2 suggest $\mathcal{B}_{\alpha,\beta}(r)$ is the set of linear models satisfying $w \leq w_*$, which implies

$$\Pr\{\mathcal{C}_{\alpha,\beta}(\mathcal{B}_{\alpha,\beta}(r))\} = \Pr\{I_\alpha^\beta(x, x'; h_*)\}. \tag{18}$$

To verify this, we first show every $(x, x') \in \mathcal{C}_{\alpha,\beta}(\mathcal{B}_{\alpha,\beta}(r))$ makes event $I_\alpha^\beta(x, x'; h_*)$ true. This is true because, for any $x, x'$ with $d(x, x') \leq \alpha$, if there exists an $w \leq w_*$ such that $|wx - wx'| > \beta$, then $|w_* x - w_* x'| \geq |wx - wx'| > \beta$. We then show every $(x, x')$ that makes event $I_\alpha^\beta(x, x'; h_*)$ true is also in $\mathcal{C}_{\alpha,\beta}(\mathcal{B}_{\alpha,\beta}(r))$. This is true since $h_*$ exists.

The equivalence implies $\xi = \sup_{r>0} \frac{\Pr\{I_\alpha^\beta(x, x'; h_*)\}}{r} = \sup_{r>0} \frac{r}{r} = 1$. The proof is completed. $\square$

## 5 IMPLEMENTATION

In this section, we discuss three implementation issues.

The first issue is related to the AMF Learner in Definition 3.4. Directly solving (7) is not easy since $\Delta_{\alpha,\beta}(h)$ is non-convex. We propose to approximate the solution by solving

$$\min_{h \in H} \frac{1}{n} \sum_{i=1}^n \ell(h(x_i), y_i) + \lambda\,\tilde{\Delta}_{\alpha,\beta}(h; S), \tag{19}$$

instead, where $\lambda$ is a regularization coefficient and

$$\tilde{\Delta}_{\alpha,\beta}(h; S) = \frac{1}{n^2 \beta^2} \sum_{i,j=1}^n M_{ij} \cdot |h(x_i) - h(x_j)|^2, \tag{20}$$

with $M$ being an $n$-by-$n$ matrix whose entries are defined as $M_{ij} = \mathbb{I}\{d(x_i, d_j) \leq \alpha\}$. Such approximation can be justified by the following relation, which implies that minimizing $\tilde{\Delta}_{\alpha,\beta}(h; S)$ also minimizes $\Delta_{\alpha,\beta}(h; S)$.

**Lemma 5.1.** *Fix any $\alpha, \beta > 0$. We have $\Delta_{\alpha,\beta}(h; S) \leq \tilde{\Delta}_{\alpha,\beta}(h; S)$ for any $h \in S$ and sample $S$.*

In practice, the approximate learner (19) may not always return a model with zero bias on training data. In this case, the proposed algorithm remains applicable and sample-efficient

on fairness. There are two possible theoretical explanations on the maintained efficiency. First, if the bias is sufficiently small e.g., $\Delta_{\alpha,\beta}(h;S) \in O(\varepsilon)$, then the passive bound in Theorem 3.6 can be extended to $\Delta_{\alpha,\beta}(h) \in O(\varepsilon)$. Plugging this back to Theorem 4.2, we can obtain a similar complexity with an additional constant factor. Second, we may borrow ideas from agnostic active learning e.g., Dasgupta et al. [2007], Balcan et al. [2009] and develop a new complexity for the non-realizable case (i.e., when $h$ has zero bias). These possible extensions are left for future study.

The second implementation issue is related to the base model. We propose to implement a linear model and a kernel regression model approximated by Random Fourier Feature Rahimi and Recht [2007] – we call it 'rff model'.

For the linear model, if instances $x_1, \ldots, x_n \in \mathbb{R}^p$, we can show $\tilde{\Delta}_{\alpha,\beta}(h;S) = \frac{2}{n^2\beta^2} \cdot h^T[x](D-M)[x]^T h$, where $[x]$ is an $n$-by-$p$ matrix with the $i_{th}$ row being $x_i^T$. Further, if squared loss is used, then solution to (19) is

$$ h = ([x](I - \frac{2\lambda}{n\beta^2}(D-M))[x]^T)^{-1}([x][y]), \quad (21) $$

where $[y] \in \mathbb{R}^n$ is a vector with the $i_{th}$ entry being $y_i$ and $D$ is an $n$-by-$n$ diagonal matrix with $D_{ii} = \sum_{j=1}^{n} M_{ij}$.

For the rff model, we first calculate random features Rahimi and Recht [2007] and then train a linear model based on them using the AMF learner. Note random features are only used to approximate the prediction model, and we still measure $d(x, x')$ using the original features.

The last issue is related to active learning. Given a labeled training set $L$ and an unlabeled set $U$, the proposed active AMF learner labels a candidate instance $u$ if there exists $u' \in L$ satisfying $d(u, u') \le \alpha$ and $|h(u) - h(u')| > \beta$. In principle, we can also pair $u$ with instances in $U$, as long as the labeled instances fall in $\mathcal{C}_{\alpha,\beta}(V_q)$. In practice, pairing $u$ with instances in $L$ is often more efficient (since the label set is often way smaller than the unlabeled set), and leads to slightly better performance as we observe in experiments.

## 6 EXPERIMENT

We experiment on three real-world data sets. The Insurance data setdat [a] has individual medical costs billed by health insurance company, and the task is to predict the cost based other attributes. The Life data setdat [b] has the life expectancy in different countries, and the task is to predict the expectancy. We also use a data set collected from public resources. It contains the COVID death rates of 3142 counties in United States and the task is to predict the rate based on other attributes including population density, obesity rate, smoking rate, diabetes rate, elderly population and vaccine rate. To learn more data sets used to evaluate algorithmic fairness, we refer interested readers to Le Quy et al. [2022].

We encode categorical features by dummy variables, address missing data using mean imputation and standardize all features. For higher numerical stability, we re-scale the labels: on the Insurance data set, we divide the medical cost which varies from 4k to 40k by 10k; on the Life data set, we divide the life expectancy which varies from 40 to 90 by 100; on the COVID data set, we multiply the death rate which varies from 0 to 0.01 by 100.

We randomly split each data set into an initial training set (assumed labeled), an unlabeled set (for query) and a testing set. Size of the initial training set is chosen as follows: for the linear base model, it is the feature number on the Insurance and Life data sets, and twice that number on the COVID data set; for the rff base model, it is half of the random feature number. Size of the testing data is 25% of the total data size. The remaining data are treated as unlabeled.

On a data set, we run an active learner for 20 random trials and report the average model performance on the testing sets. Model bias is measured by $\Delta_{\alpha,\beta}(h; S_n)$ defined in (6), with $(\alpha, \beta)$ set to (2, 0.1) on Insurance, (10, 0.2) on Life and (1.5, 0.001) on COVID. We also experimented with other fairness coefficients and observed similar comparative performance. Model error is measured by the root-mean-squared-error.

We evaluate the proposed active labeling strategy on the linear base model and rff base model respectively, and compare its performance with the following three strategies.

– *Random*: It randomly selects instances to label.

– *Query-by-Committee (QBC)*: It labels instances which receive the largest prediction variance from a committee of models. Following Burbidge et al. [2007], we construct a committee of five models and train each one using a bootstrap sample of the training data, with sample size equals to the training set size divided by the committee size.

– *Uncertainty*: It labels instances which are most different from the training data in both feature space and label space Wu et al. [2019]. To our knowledge, this is a state-of-the-art active labeling method for regression model.

– *Cluster*: It is a clustering based baseline method that relies on the distance between instances. It first identifies the top $m$ uncertain instances in the candidate pool using the above method, then runs k-means clustering to identify their $k$ centers, and finally labels the identified instances.

For the metric-fair learner, we pick its regularization coefficient $\lambda$ that strikes a good balance between fairness and accuracy. For the linear base model, $\lambda$ is set to 1 on Insurance and Life and 0.1 on COVID; for the rff base model, $\lambda$ is set to 1 on Insurance, 5 on Life and 0.5 on COVID.

For the rff base model, we generate the random features that approximate Gaussian kernel Rahimi and Recht [2007]. The random feature number is set to 100 on Insurance, 400 on Life and 200 on COVID. The gamma coefficient is set

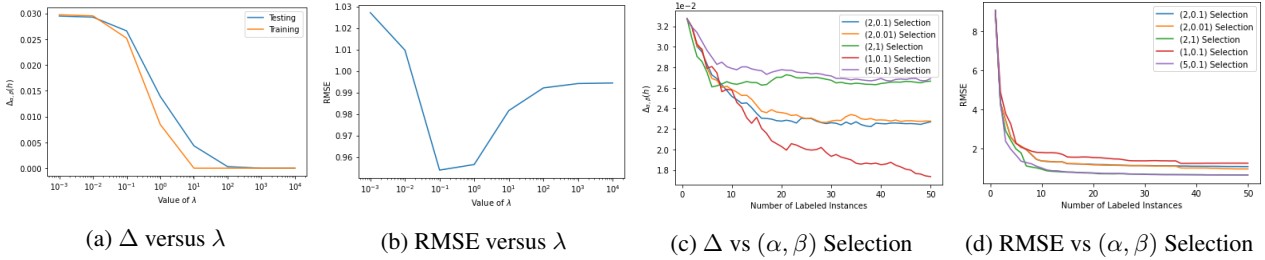

Figure 2: Results of Sensitivity Analysis

to 1e-4 on Insurance, 1e-9 on Life and 1e-2 on COVID. In practice, we observe these configurations lead to good and stable performance of active metric-fair learning. For the clustering based baseline method, we set $m = 10$ and $k = 3$ as they give consistently good performance (except $k$ is set to 10 for linear model on the Life dataset).

### 6.1 RESULTS AND DISCUSSIONS

Results of the experimented strategies on both base models across three data sets are shown in Figure 3.

In Figure 3 (a-f), we see the proposed active AMF learner reduces model bias more efficiently than other learners, which empirically verifies its efficient sample complexity. We notice it achieves almost zero bias in all cases, supporting our assumption on the realizable case. (And note this is not achieved at the cost of significantly deteriorating accuracy, as explained in the next paragraph.) There seems no consistent pattern on the efficiency of other learners. We notice QBC and uncertainty are often less efficient than random, implying the importance of (efficiently) achieving individual fairness by design, as presented in this study.

In Figure 3 (g-l), it is not surprising to see that uncertainty based labeling reduces error faster than other strategies. Comparatively, the proposed active AMF learner manages to achieve a comparable reduction rate, suggesting its has an efficient fairness-accuracy trade-off.

We also perform sensitivity analysis on the proposed strategy and present results in Figure 2. Figures 2 (a-b) show the performance versus regularization coefficient $\lambda$. We both training and testing $\delta$ decrease as $\lambda$ increases. This suggests the metric-fair learner can effectively reduce bias and the reduction is generalizable, which supports Theorem 3.6. We also see model error first decreases and then increases, exhibiting an overfitting phenomenon.

Figures 2 (c-d) show the performance versus different choices of $(\alpha, \beta)$ when selecting instances in Step 3 of Algorithm 1. (But all $\delta$'s are evaluated based on the same $(\alpha, \beta)$ for fair comparison.) We see using smaller $\alpha$ to select instances leads to faster convergence of $\delta$ but more slowly convergence of RMSE. There seems no clear pattern on the

impact of $\beta$. Overall, we see one can balance fairness and accuracy of the proposed strategy through adjusting $\alpha$.

## 7 CONCLUSION

In this paper, we propose the first active approximate metric-fair (AMF) learner and prove it can achieve an $\varepsilon$ bias budget by labeling only $O(\log \frac{1}{\varepsilon})$ instances. To our knowledge, this result is a first and substantial improvement over the existing $O(\frac{1}{\varepsilon^2})$ sample complexity for achieving individual fairness by the passive learners. Through extensive experiments across three public data sets, we show the proposed active AMF learner improves fairness of two regression models more efficiently than its passive counterpart as well as state-of-the-art active learners, while being able to maintain comparable accuracy. Another contribution of this study is to present a provably equivalent form of AMF based on uniform continuity instead of the existing almost Lipschitz.

### Acknowledgements

We thank all anonymous reviewers for their insightful feedback for enhancing the quality of the paper. We also thank Skyler Sprecker for sharing the COVID data set. This paper is based upon work supported by the US National Science Foundation under Grant No. 2101936.

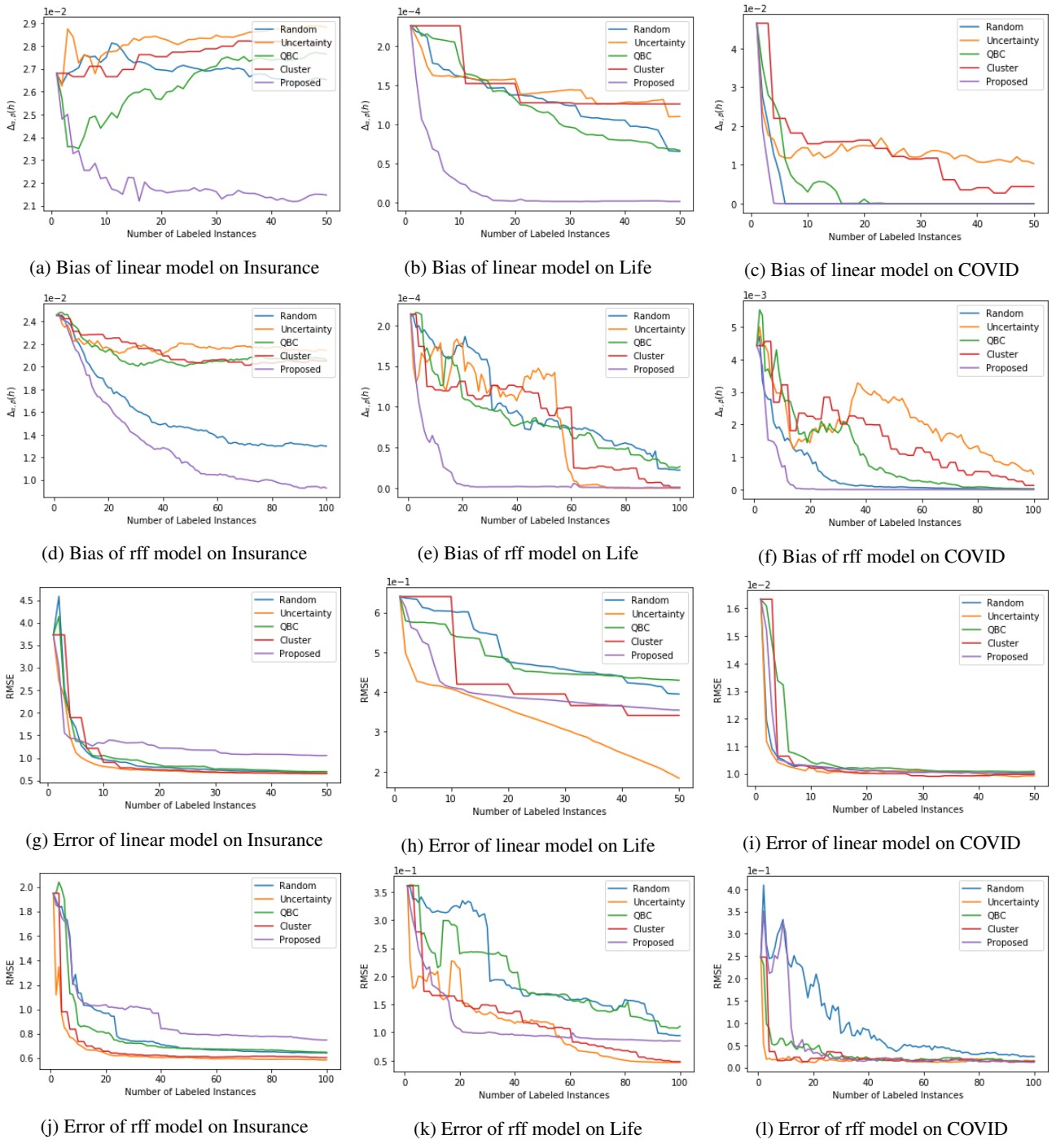

Figure 3: Performance of Different Active Labeling Strategies on Three Data Sets. (a-c) and (d-f) show the bias of linear and rff base models respectively; (g-i) and (j-l) show the rmse of linear and rff base models respectively.

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
