# OpenReview forum: "Active Approximately Metric-Fair Learning"
_auai.org/UAI/2022/Conference — UAI 2022 Poster_

### Official Review · Reviewer_DBgf · 2022-04-10

**Q2(1) Originality/Novelty:** 3
**Q2(2) Significance/Impact:** 3
**Q2(3) Correctness/Technical Quality:** 3
**Q2(6) Clarity Of Writing:** 3
**Q6 Overall Score:** 5
**Q8 Confidence In Your Score:** 3

**Q1 Summary And Contributions:**

This paper discusses a new active learning algorithm for individual fairness. The introduced algorithm can reduce the number of labeled instances from $\mathcal{O}(\frac{1}{\varepsilon^2})$ to $\mathcal{O}(\log\frac{1}{\varepsilon})$. Experiments demonstrate the theoretical findings.

**Q2 Assessment Of The Paper:**

More detailed information regarding each of these aspects is given below:

**Q2(4) Quality Of Experiments (Optional):**

3: Good: The experimental evaluation is adequate, and the results convincingly support the main claims.

**Q2(5) Reproducibility:**

2: Fair: Key resources (e.g., proofs, code, data) are unavailable but key details (e.g., proof sketches, experimental setup) are sufficiently well-described for an expert to confidently reproduce the main results.

**Q3 Main Strengths:**

1. The proposed algorithm is simple and useful.
2. There are good theoretical guarantees for the proposed algorithm.

**Q4 Main Weakness:**

1. It would be great if the authors can clearly point out which parts mostly help to reduce the number of labeled instances from $\mathcal{O}(\frac{1}{\varepsilon^2})$ to $\mathcal{O}(\log\frac{1}{\varepsilon})$. To my understanding, there are three things to make that happen: the labeling strategy in step 3 in Algorithm 1, the requirement of the number of labeled instances $k$ in each epoch, and an assumption on the counter coefficient. Adding a discussion to show readers what contributes to the reduction would be super helpful.
2. There is no caption for Fig. 1 which brings confusion. Also, the caption for Fig. 2 is not informative, making the figure hard to understand.

**Q5 Detailed Comments To The Authors:**

1. Probably for Theorem. 3.6 it would be necessary to interpret the variable $t$ since $t$ is introducing additional errors into $\Delta_{\alpha,\beta}$ and affects $m$.
2. It seems that we must find and have $k$ newly labeled instances which satisfy Eq. 12. Does this mean the algorithm is introducing new requirements on the unlabeled set $U$?
3. In Algorithm 1 step 2, it is unlikely that a model training can perfectly satisfy Definition 3.4. Does the algorithm still work if there is a violation on Eq. 7 when doing step 2?

Minor:
1. In the proof sketch of Theorem 3.6, what is $d(\alpha,\beta)$?
2. In the main paper, Theorem A.2 -> Theorem 3.6?


**Q7 Justification For Your Score:**

Overall I think this is a good paper. The research is well-motivated, and the proposed algorithm is theoretically and technically sound. Reducing the number of labeled instances is meaningful. I will definitely raise my score if the authors can address my concerns.

**Q9 Complying With Reviewing Instructions:**

1: Yes.

---

### Official Review · Reviewer_RUJX · 2022-04-11

**Q2(1) Originality/Novelty:** 2
**Q2(2) Significance/Impact:** 2
**Q2(3) Correctness/Technical Quality:** 2
**Q2(6) Clarity Of Writing:** 2
**Q6 Overall Score:** 4
**Q8 Confidence In Your Score:** 3

**Q1 Summary And Contributions:**

This paper proposes an active approximate metric-fair (AMF) learner built upon the original form of AMF and theoretically prove it can achieve the same budget with less labeled instances.

**Q2 Assessment Of The Paper:**

More detailed information regarding each of these aspects is given below:

**Q2(4) Quality Of Experiments (Optional):**

1: Poor: The experimental evaluation is flawed or the results fail to adequately support the main claims.

**Q2(5) Reproducibility:**

2: Fair: Key resources (e.g., proofs, code, data) are unavailable but key details (e.g., proof sketches, experimental setup) are sufficiently well-described for an expert to confidently reproduce the main results.

**Q3 Main Strengths:**

1. The research question is new and interesting.
2. The paper is easy to follow.

**Q4 Main Weakness:**

1. It is unclear to me that how individual fairness is achieved in Definition 3.2 while the original AMF in Definition 3.1 is straightforward. Why Definition 3.2 enforce the idea of similar individual being treated similarly? Also, the authors motivate Definition 3.2 from Definition 3.1 simply as 'To facilitate algorithm design and analysis' which are unclear to me as well. More explanation is expected.

2. Relevantly, the main contributions of this paper remain unknown to me which seems like an extension of AMF to active settings.

3. The higher sample efficiency is proved theoretically but not experimentally. It is also questionable that the experiments are extensive but are limited instead.



**Q5 Detailed Comments To The Authors:**

More experiments are expected considering SOTA, e.g., "Fair Active Learning", to justify the use the datasets, e.g., "A survey on datasets for fairness‐aware machine learning", and to verify its claimed efficiency.

**Q7 Justification For Your Score:**

See Q3 & Q4

**Q9 Complying With Reviewing Instructions:**

1: Yes.

---

### Official Review · Reviewer_rs1P · 2022-04-13

**Q2(1) Originality/Novelty:** 3
**Q2(2) Significance/Impact:** 3
**Q2(3) Correctness/Technical Quality:** 3
**Q2(6) Clarity Of Writing:** 4
**Q6 Overall Score:** 7
**Q8 Confidence In Your Score:** 3

**Q1 Summary And Contributions:**

The paper proposes an active learning framework to obtain *individual* fairness in regression settings in O(log 1/\epsilon) rather than O(1/\epsilon^2) samples, while maintaining reasonable accuracy.

**Q10 Ethical Concerns (Optional):**

N/A. The whole point of the paper is to take bias/ethical concerns into account.

**Q2 Assessment Of The Paper:**

More detailed information regarding each of these aspects is given below:

**Q2(4) Quality Of Experiments (Optional):**

4: Excellent: The experimental evaluation is comprehensive and the results are compelling.

**Q2(5) Reproducibility:**

3: Good: Key resources (e.g., proofs, code, data) are available and key details (e.g., proofs, experimental setup) are sufficiently well-described for competent researchers to confidently reproduce the main results.

**Q3 Main Strengths:**

- I found the paper well-written and easy to follow, and the comparison with previous work clear
- The main result is compelling. The authors greatly reduce the sample complexity needed (from quadratic to logarithmic) by using an active learning approach (rather than sampling i.i.d. and using standard Rademacher complexity techniques).
- The idea of how the algorithm works is also interesting. At each time step, the authors reduce the probability mass/volume of the hypothesis space by 1/2, which shows in turn that only logarithmically many steps are needed.
- The authors provide some calculations for the counter AMF coefficient in a few simple settings, showing how to obtain it in practice.
- The experiments are extensive, with the authors comparing their results to the state-of-the-art on three datasets, both for linear regression and kernel regression. They show  significant bias reduction at a faster rate compared to SOTA, with a relatively small costs in terms of accuracy.

**Q4 Main Weakness:**

The main weakness of the paper is that it is hard to understand what the AMF coefficient looks like:
- The authors provide examples in 1D settings of how to obtain the coefficient and what its value is, but what happens in high-dimensional settings? How does the AMF factor scale?
- The examples they provide rely on a restricted assumption that instances are uniformly distributed. I think it would be nice to look at more realistic population distributions (e.g., Gaussians)
- Since the results rely on this AMF factor being finite, it would be useful to provide conditions under which this is the case.
- A lot of the work on fairness in such online settings expresses their results in terms of regret bounds instead of sample complexity. It would be good if the authors could translate their current sample complexity bound into a regret bound on the bias for the sake of comparison. Since they obtain \exp(-T) accuracy after T rounds, these bounds should be competitive with previous work.

**Q5 Detailed Comments To The Authors:**

See above for detailed comments and questions.

**Q7 Justification For Your Score:**

I think this is overall a strong paper with good theoretical results, and nice experiments to validate these theoretical results. The only thing that I feel is missing from the paper is a better understanding of the counter AMF coefficient and conditions on when it is finite or relatively small, as well as dependencies on the dimension of the problem. Overall, I think the paper should be accepted at UAI.

**Q9 Complying With Reviewing Instructions:**

1: Yes.

---

### Official Review · Reviewer_wkC8 · 2022-04-16

**Q2(1) Originality/Novelty:** 3
**Q2(2) Significance/Impact:** 2
**Q2(3) Correctness/Technical Quality:** 2
**Q2(6) Clarity Of Writing:** 3
**Q6 Overall Score:** 5
**Q8 Confidence In Your Score:** 3

**Q1 Summary And Contributions:**

Existing methods for epsilon-bias budget need O(1/\epsilon^2) samples and they are usually passive. This paper proposes an approximate metric learning framework (AMF) that can achieve epsilon-budget using only O(log(1/\epsilon)) instances. This is a huge increase in the sample complexity. The main idea is to select samples that are close to each other with different model outputs as shown in Algorithm 1 in the paper.


**Q2 Assessment Of The Paper:**

More detailed information regarding each of these aspects is given below:

**Q2(4) Quality Of Experiments (Optional):**

2: Fair: The experimental evaluation is weak: important baselines are missing, or the results do not adequately support the main claims.

**Q2(5) Reproducibility:**

3: Good: Key resources (e.g., proofs, code, data) are available and key details (e.g., proofs, experimental setup) are sufficiently well-described for competent researchers to confidently reproduce the main results.

**Q3 Main Strengths:**

The paper is well written and addresses an important problem dealing with individual fairness. The problem is still relatively novel, and there are only a few methods out there to ensure approximate metric fairness.

The theoretical sections of the paper are well written with formal statements and proofs, and also providing the necessary intuitions.

The actual Algorithm 1 for AMF learning is simple.


**Q4 Main Weakness:**

The experimental section can be strengthened a bit more. First of all, the baselines are either uniform or uncertainty based. Since the AMF learning involves considering points that are close and have different model predictions, it would be interesting to use baselines that rely on the distance d(x,x’) between pairs of samples x and x’. Some possibilities include k-center or submodular approaches with potential constraints to enforce fairness.


Although AMF seems to do well on the fairness metric, it appears that uncertainty-based sampling consistently outperforms the proposed algorithm by a large margin on RMSE. It would be good to show some tradeoff between the two metrics through appropriate choices of parameters.


Step 3 in algorithm 1 needs more analysis. What is the complexity of finding points that satisfy conditions shown in Equation (12). For large datasets, this can be a serious issue.


**Q5 Detailed Comments To The Authors:**

See my comments above.

**Q7 Justification For Your Score:**

Please address my concerns above.

**Q9 Complying With Reviewing Instructions:**

1: Yes.

---

### Decision · Program_Chairs · 2022-05-15

**Decision:**

Accept (Poster)

**Comment:**

Meta Review: This paper proposes an active approximately metric-fair (AMF) learner that improves the sample efficiency from O(1/\epsilon^2) in the passive setting to O(log 1/\epsilon). In the AC’s opinion, this is a huge improvement and offers a very compelling alternative to the passive solution. This is acknowledged and highlighted by reviewers too. The above improvement is achieved with maintained and reasonable accuracy.